# Assessment of treatment-specific tethering survival bias for the juvenile blue crab *Callinectes sapidus* in a simulated salt marsh

Cole R. Miller[1,2], A. Challen Hyman[2]*, Daniel H. Shi[1], Romuald N. Lipcius[2]*

**1** William & Mary, Williamsburg, Virginia, United States of America, **2** Virginia Institute of Marine Science, William & Mary, Gloucester Point, Virginia, United States of America

☉ These authors contributed equally to this work.
* achyman@usf.edu (ACH); rom@vims.edu, rnlipc@wm.edu (RNL)

**Data Availability Statement:** All relevant data are within the paper and its Supporting information files.

## Abstract

The blue crab (*Callinectes sapidus*) is ecologically and economically important in Chesapeake Bay. Nursery habitats, such as seagrass beds, disproportionately contribute individuals to the adult segment of populations. Salt marshes dominated by smooth cordgrass *Spartina alterniflora* are intertidal nursery habitats which may serve as a refuge from predation for juvenile blue crabs. However, the effects of various characteristics of salt marshes on nursery metrics, such as survival, have not been quantified. Comparisons of juvenile survival between salt marshes and other habitats often employ tethering to assess survival. Although experimental bias when tethering juvenile prey is well recognized, the potential for habitat-specific bias in salt marshes has not been experimentally tested. Using short-term mesocosm predation experiments, we tested if tethering in simulated salt marsh habitats produces a habitat-specific bias. Juvenile crabs were tethered or un-tethered and randomly allocated to mesocosms at varying simulated shoot densities and unstructured sand. Tethering reduced survival, and its effect was not habitat specific, irrespective of shoot density, as evidenced by a non-significant interaction effect between tethering treatment and habitat. Thus, tethering juvenile blue crabs in salt marsh habitat did not produce treatment-specific bias relative to unvegetated habitat across a range of shoot densities; survival of tethered and un-tethered crabs was positively related to shoot density. These findings indicate that tethering is a useful method for assessing survival in salt marshes, as with other nursery habitats including seagrass beds, algae and unstructured sand.

## Introduction

A major objective in estuarine ecology and fisheries science is understanding the influence of habitat on the population dynamics of commercially exploited species. in particular, ecologists have emphasized identifying critical juvenile habitats largely because early life stages are the most vulnerable for fish and aquatic invertebrates [1–3]. As the extent and quality of productive coastal habitats is deteriorating [4–6], information on highly productive habitats is particularly valuable for conservation and management [7, 8].

**Funding:** Preparation of this manuscript was funded by a Willard A. Van Engel Fellowship of the 295 Virginia Institute of Marine Science, William & Mary, the NMFS-Sea Grant Joint 296 Fellowship 2021 Program in Population and Ecosystem Dynamics, and the National 297 Science Foundation (grant number NSF OCE 1950242 to R. D. Seitz).

**Competing interests:** The authors have declared that no competing interests exist.

Habitat-specific juvenile survival rates are an important attribute of productive habitats [1, 9, 10]. As survival rates are especially low during early juvenile stages, habitat-specific differences in survival rates can lead to substantial variation in secondary production of juveniles and affect population dynamics at large spatial scales [11]. Hence, quantifying habitat-specific and life-history stage survival rates are vital to properly estimate the function and value of estuarine habitats.

Tethering–a method that restrains a prey to a fixed location for a period of time–is commonly utilized to quantify relative survival rates in field experiments. Tethering allows a researcher to be absent during field trials, thereby reducing the probability of unnatural predator and prey behavior induced by humans [12, 13]. Tethering in field settings also provides prey with a wider range of movement and the ability to feed and hide as opposed to alternative methods such as caging [14, 15]. However, tethered animals are more vulnerable to predation due to restricted movement and altered escape behaviors which may artificially reduce survival [13]. Hence, this method can bias survival rates. Consequently, predation rates from tethering are interpreted as relative, not absolute [15]. Moreover, a major assumption that enables comparison of survival across habitats is that the bias associated with the tethering process is consistent across all treatments [13]. Thus, it is imperative that treatment-specific biases such as the interactions between characteristics of the habitats and the tethering procedure be addressed prior to any survival study utilizing tethering [12, 13].

The blue crab, *Callinectes sapidus*, is a widely distributed marine and estuarine species. Within Chesapeake Bay, the blue crab is a dominant benthic predator, which may exert top-down control on marine invertebrate communities [16], as well as a valuable food source for many commercially important species [17–20]. The blue crab supports one of the most valuable fisheries in the Western Atlantic and Gulf of Mexico; in 2019 US annual commercial landings of blue crab were 66,497 mt valued at US $205.6 million [21]. However, the blue crab population in Chesapeake Bay has fluctuated over the past two decades, reaching an all-time low of 227 million estimated crabs in 2022, the lowest abundance since the Blue Crab Winter Dredge Survey's conception in 1990 [22].

Blue crabs exploit numerous habitats in their early life stages, such as seagrass (e.g. eelgrass *Zostera marina*) and algal (*Gracilaria vermiculophylla*) beds, *Spartina alterniflora* salt marshes, and coarse woody debris [16]. Previous studies estimating survival in small juveniles (i.e. 10–35 mm) have primarily focused on seagrass meadows, algae and unstructured sand habitats to infer nursery status [23–29]. However, mounting evidence suggests that alternative habitats such as salt marshes serve as highly productive nurseries [30, 31]. While past tethering experiments have not found treatment-specific interactions among unstructured sand and seagrass meadows [25, 26, 29], such interactions have not been quantified in salt marsh nurseries [32]. Given marked structural differences between *S. alterniflora* and *Z. marina* vegetation, assuming that treatment-specific bias is negligible in salt marshes based on tests of treatment-specific bias in seagrass beds may lead to spurious inferences on *S. alterniflora* refuge quality.

In this study, we examined survival of tethered and untethered juvenile blue crabs in artificial salt marsh and unstructured sand habitats with a series of predator-prey mesocosm experiments. Our objective was to determine whether treatment-specific bias was present among unstructured sand and salt marsh habitats. Moreover, salt marshes are not homogeneous with respect to structural complexity. Thus, we also tested for treatment-specific bias across a range of salt marsh shoot densities observed in the field.

## Logical framework

Under an information theoretic framework [33] we developed multiple alternative hypotheses ($H_i$) [34] for our short-term predation experiments. Herein we describe and justify the hypotheses and corresponding independent variables.

$H_1$: Tethering reduces survival, as restraining crabs should increase vulnerability to predation [25, 26, 29].

$H_2$: Survival increases with marsh shoot density, based on reduced encounter rates or capture efficiency for predators in structurally complex habitats [32].

$H_3$: Survival is a function of tethering status and shoot density. The effects of shoot density and tethering status may affect survival additively ($H_{3a}$). Alternatively, survival may be a function of tethering status, marsh shoot density, and their interaction effect ($H_{3b}$). For example, juvenile blue crabs may have different escape strategies under differing habitat conditions, such as crypsis in structured habitat vs. escape in unstructured habitat [13]. Tethering juveniles under different shoot densities may result in non-additive effects on survival (i.e., treatment-specific bias).

$H_4$: Survival is a function of tethering, shoot density, and predator size, such that survival is related to predator size because larger predators are restricted in movement, compared to smaller adults who can navigate through restricted spaces. As a result, large predators are less efficient than smaller predators when foraging for small juvenile crabs [32, 35–37]. $H_4$ considered both the effect of predator size within an additive model ($H_{4a}$) as well as with the effect of predator size plus a tethering-shoot density interaction (i.e., treatment-specific bias; $H_{4b}$).

$H_5$: Survival is a function of tethering, shoot density, predator size, and prey size. As juvenile blue crabs grow, they are less susceptible to predation as their carapace widens and becomes harder, spines become more prominent, and aggressive behavior intensifies [16, 26, 38, 39]. $H_5$ also considered both an additive model ($H_{5a}$) and another with the tethering-shoot density interaction ($H_{5b}$).

## Methods

### Ethics statement

All animals used in this study were low-level invertebrates that were not cephalopods and thus do not require an IACUC permit. The sacrificed animals (juvenile blue crabs) referenced in this study were euthanized humanely using the spike method, which entailed using a sharp object to quickly and effectively destroy the ganglia and central nervous system of the juvenile to prevent any suffering.

### Experimental design

The experiment employed eight 160-L recirculating cylindrical fiberglass tanks with a bottom area of 0.36 $m^2$, aerated by an air-stone, to simulate an estuarine marsh. Experimental replication was facilitated through the use of multiple trials. Each trial consisted of eight predator-prey observations within separate tanks—four tanks (one for each shoot-density treatment, see below) receiving tethered juvenile blue crabs and four receiving free-swimming juveniles (Fig 1).

Marsh grass densities were simulated using wooden dowels (1 cm diameter, 30.8 cm height) placed into a 40.6 cm x 40.6 cm plastic pegboard, which was buried 3–5 cm beneath sand from the York River. The wooden dowels were slightly smaller than the diameter of the holes within each pegboard, giving dowels a small range of motion within the holes and resembling flexibility observed in salt marsh vegetation. Adult crabs were also observed successfully navigating

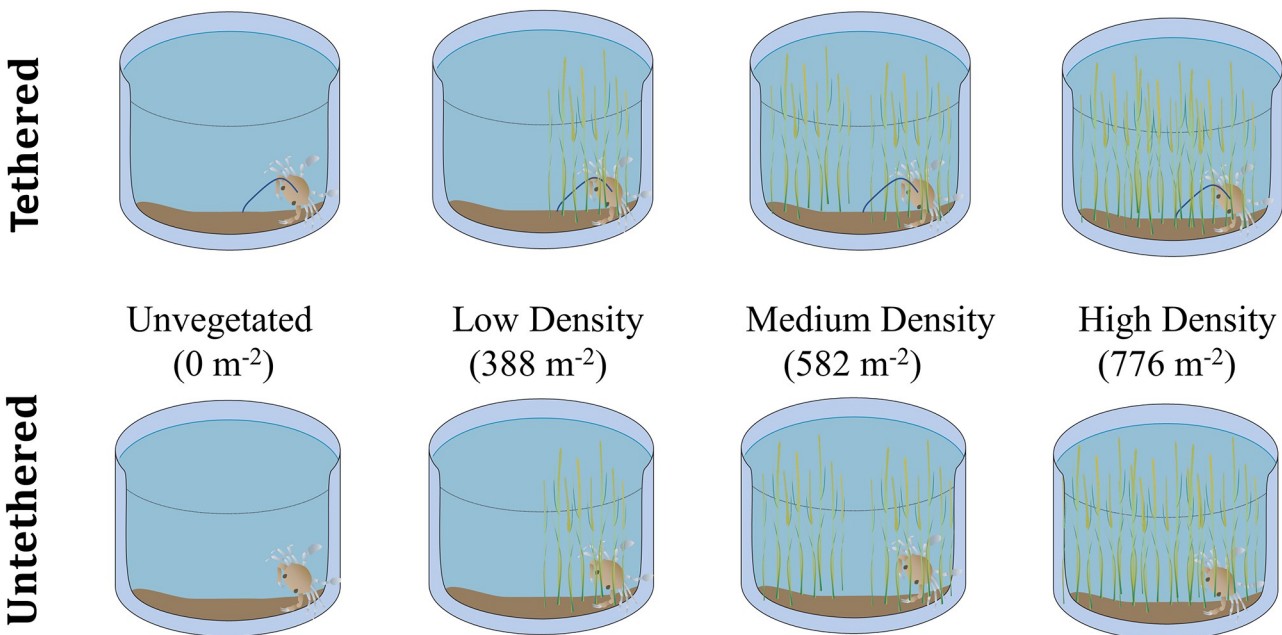

**Fig 1. Conceptual diagram depicting mesocosm experimental design and idealized tank setup for tethering-shoot density setup.** Both shoot density and tethering treatments were randomized regularly to avoid tank-specific bias.

within the dowels. Additionally, dowels were left within the tank for two weeks before the experiment start, allowing them to become waterlogged and fouled, which also increased flexibility. A control treatment included a buried peg board without dowels. Juvenile blue crabs were caught weekly using dip-nets in local seagrass and algal habitats. Prior to the experiment, juvenile blue crabs were set into tanks without a predator and recovered 24 h later after draining the tank completely (n = 14). All animals were found within 5 min of searching, which validated the assumption that missing animals after 24 hours were eaten. Four shoot density treatments with 0, 64, 96, and 128 (corresponding to densities of 0, 388, 582, and 776 m$^{-2}$, respectively) were employed based on densities in *S. alterniflora* salt marshes [40–42]. Average spacing between the shoots was 5.1, 4.14, and 3.6 cm for the 388, 582, and 776 density treatments respectively. in each tank, PVC aqueducts continuously supplied river-sourced water at a constant flow rate. Water level was controlled using a PVC standpipe in the center of each tank (Fig 2). Tank water was changed completely after each trial (i.e. every 48 h) to reduce buildup of ammonia, nitrates, and other waste compounds. Before each trial, temperature, salinity, and DO were measured in each tank with a YSI data sonde to account for natural fluctuations in the river-sourced water.

Adult blue crabs were selected as model predators, as adult conspecifics are among the most important predators of small juveniles [16, 38, 43–46]. Adult crabs were caught using crab traps in the York River. Following capture, each crab was measured, tagged, placed in a holding tank and fed juvenile blue crabs to acclimate predators to prey. Holding tanks employed the same flow-through system as experimental tanks. If an adult crab molted, it was allowed to harden for at least 2 d prior to a trial. Adult crabs were acclimated to experimental conditions, in separate cages to deter antagonistic behavior, for 14 d prior to their first trial. Juvenile blue crabs (prey) were acclimated in a third, separate tank system with individual compartments to deter antagonistic behavior.

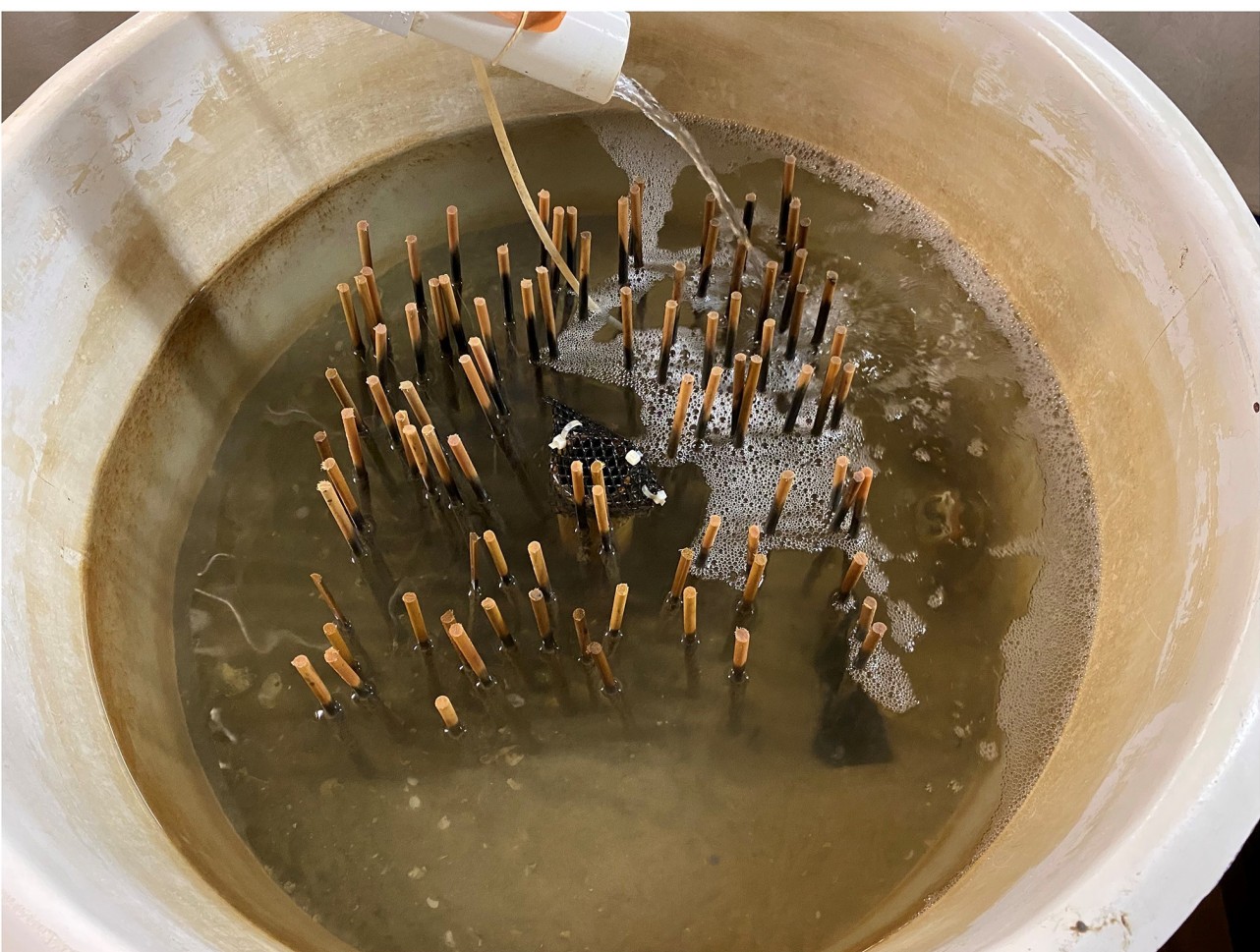

**Fig 2. Photograph of one of the replicate tanks (96 shoots tank$^{-1}$ or 582 shoots m$^{-2}$) and flow-through system.** All tanks had an identical setup except for differences among density treatments. One of the predators is visible on the right side of the tank.

Eight juvenile crabs (four tethered, four untethered) were randomly selected prior to each trial. Tethers were applied to all juveniles assigned to the tethering treatment 24 hours before each trial. To adhere the tether, cyanoacrylate glue (super glue) was applied to the dorsal side of the carapace of juveniles between the anterior pair of walking legs to not inhibit movement. A 25 cm thread of monofilament fishing line (4.5 kg test) was set into the glue [25, 27] with the length of the line allowing the juveniles to navigate the entirety of the tank. A small square of brown duct tape was then applied over the thread and glue to secure the glue-filament base. After tethering, each prey was moved into a holding tank to acclimate before the start of a trial. For each

**Table 1. Summary tables describing shoot density-tethering treatment replicates.**

| Tether treatment | Shoot density treatment | | | |
|---|---|---|---|---|
| | **0** | **388** | **582** | **776** |
| Untethered | 10 | 8 | 8 | 8 |
| Tethered | 9 | 9 | 9 | 9 |
| Total | 19 | 17 | 17 | 17 |

trial, one juvenile blue crab was randomly assigned to a single replicate in one of the two tethering treatments (tethered or untethered), with a given replicate consisting of one of four density treatments, resulting in 8 tanks per trial (n = 2 tether treatments x 4 shoot density treatments = 8). Water depth was held constant at 24 cm, simulating high tide conditions in local salt marshes. Every two weeks, shoot density treatments were randomized to control for tank effects. This experiment was replicated for 11 trials (n = 8 tanks x 11 trials = 88, Fig 1 and Table 1).

For each trial replicate, an adult blue crab (predator) was selected randomly from the holding tank and placed into each experimental tank. After recording its carapace width, a juvenile blue crab was placed in the center of each tank near the PVC standpipe regardless of treatment. The weight at the end of the tether was similarly placed at the center of the tank. Each trial ran for 24 h, after which the adult crab was removed with a dip net, placed in a separate tank, fed, and left to reacclimate for 24 h before the next trial. Subsequently, each tank was completely drained via a plastic siphon and searched for 10 min (i.e. twice the duration estimated to recover surviving juveniles) for surviving juveniles or carapace fragments. The absence of a juvenile crab or presence of carapace fragments was interpreted as a predation event. Then, tanks were refilled. If an adult crab behaved abnormally, another adult was selected randomly from a holding tank containing replacement crabs.

## Data analysis

At the conclusion of each experiment, binary survival data were analyzed using generalized linear regression mixed-effects models to evaluate effects of experimental treatments and water chemistry variables. The response variable (probability of juvenile survival) was modeled using a binomial distribution and related to predictor variables using the logit-link (i.e. logistic regression). Predator ID, trial number, and tank ID were initially included as random-intercept effects but discarded due to negligible residual variation explained in all cases, which reduced the model structures to generalized linear models.

The hypotheses for each experiment were translated to sets of statistical models ($g_i$; Table 2) and evaluated within an information theoretic framework [33, 47]. Salinity, temperature, and dissolved oxygen (DO) were initially included as fixed effects to ensure that variation in these variables did not influence survival, and were subsequently eliminated from consideration. For each model set, AIC (Akaike's information Criterion) corrected for small sample size (AICc) was employed to evaluate the degree of statistical support for each model. Weighted model probabilities ($w_i$) based on $\Delta_i$ values were used to determine the probability that a particular

**Table 2. Information theoretic analysis of 8 logistic regression models ($g_i$) using tethering status (T), shoot density (SD), predator size (P), and prey size (Sp) as predictors of juvenile blue crab survival, where $AIC_c$ is the Akaike information criterion corrected for small sample size, $\Delta_i$ is the difference between any model and the best model in the set, and $w_i$ is the weighted model probability that a given model is the best among the set considered. Values from the selected model are presented in bold font.**

| Hypothesis | Model: Formula | $AIC_c$ | $\Delta_i$ | $w_i$ |
|---|---|---|---|---|
| $H_1$ | $g_1$: T | 97.23 | 4.59 | 0.03 |
| $H_2$ | $g_2$: SD | 95.08 | 2.44 | 0.09 |
| $H_{3a}$ | **$g_3$: T + SD** | **92.64** | **0.00** | **0.30** |
| $H_{3b}$ | $g_4$: T + SD + (T x SD) | 94.62 | 1.98 | 0.11 |
| $H_{4a}$ | $g_5$: T + SD + P | 92.93 | 0.29 | 0.26 |
| $H_{4b}$ | $g_6$: T + SD + (T x SD) + P | 95.23 | 2.59 | 0.08 |
| $H_{5a}$ | $g_7$: T + SD + P + Sp | 94.74 | 2.10 | 0.10 |
| $H_{5b}$ | $g_8$: T + SD + (T x SD) + P + Sp | 97.13 | 4.49 | 0.03 |

model was the best-fitting model in a set. Models with $\Delta_i$ values within two points of the best fitting model were considered to have comparable support and further evaluated using likelihood ratio $X^2$ tests to determine their importance [33, 48]. When two models had comparable $\Delta_i$ values and likelihood ratio $X^2$ tests did not suggest significant differences in explanatory power, the simpler model was chosen as the more appropriate model under the principle of parsimony.

## Results

A total of 88 tank-trial combinations were run between June and July 2022, although only 70 were used due to various logistical issues (e.g. incorrect data sonde readings). Data and ranges for physicochemical variables (DO, temperature, and salinity) and sizes of prey and predators are detailed in Table 3 and Fig 2.

### Model selection

The best model was $g_3$, an additive model with tethering and shoot density as its predictors (Table 2). All other models except $g_4$ and $g_5$ were eliminated from consideration because their weighted probabilities were less than or equal to 0.1. Although models $g_3$ and $g_5$ had similar AICc values, model $g_5$ did not improve predictive performance (likelihood ratio $X^2$ test, p = 0.16) and the added parameter in model $g_5$, predator width, was not statistically significant (p = 0.17). Hence, we selected model $g_3$ as the most appropriate model.

We also compared $g_3$ and $g_4$ ($g_3$'s companion model which included the shoot density-tethering interaction term) to test for treatment-specific bias. Model $g_4$ with one more parameter did not improve fit to the data (likelihood ratio $X^2$ test, p = 0.60) and the interaction effect was not statistically significant (p = 0.61). Thus, we conclude that treatment-specific bias was absent.

in model $g_3$, survival increased with shoot density, such that for every unit (1 shoot) of shoot density, the odds of survival increased by 0.23% (Fig 3). in contrast, tethered crabs were significantly less likely to survive by 111% relative to free-swimming crabs (Table 4).

## Discussion

The major findings of this study were that (1) tethering had a significant negative effect on survival, and (2) treatment-specific tethering bias across habitats was not present. The negative effect of tethering was consistent with literature [13–15, 26] and supports the assertion that tethering results in salt marsh habitats are a reliable, relative measure of habitat-specific survival. in addition, there was no significant interaction effect between tethering and shoot density on crab survival, demonstrating that treatment-specific bias was not present irrespective of shoot density. Taken together with earlier studies reporting no treatment-specific tethering bias among seagrass meadows and unstructured sand habitats [25, 29], our findings expand the portfolio of habitats under which tethering may be reliably employed with respect to juvenile blue crabs. A benefit of the present study is that results presented here retroactively

**Table 3. Summary statistics for physicochemical and biological variables Salinity (Sal), Temperature (Temp), Dissolved Oxygen (DO), Predator Width (Predator CW), and Prey Width (Prey CW).** CW = carapace width.

| Estimate | Sal | Temp (°C) | DO (mg/L) | Predator CW (mm) | Prey CW (mm) |
|---|---|---|---|---|---|
| Minimum | 16.91 | 23.60 | 5.60 | 104.00 | 11.30 |
| Mean | 19.15 | 24.97 | 6.38 | 132.21 | 20.24 |
| Maximum | 20.35 | 26.20 | 7.75 | 155.00 | 30.20 |

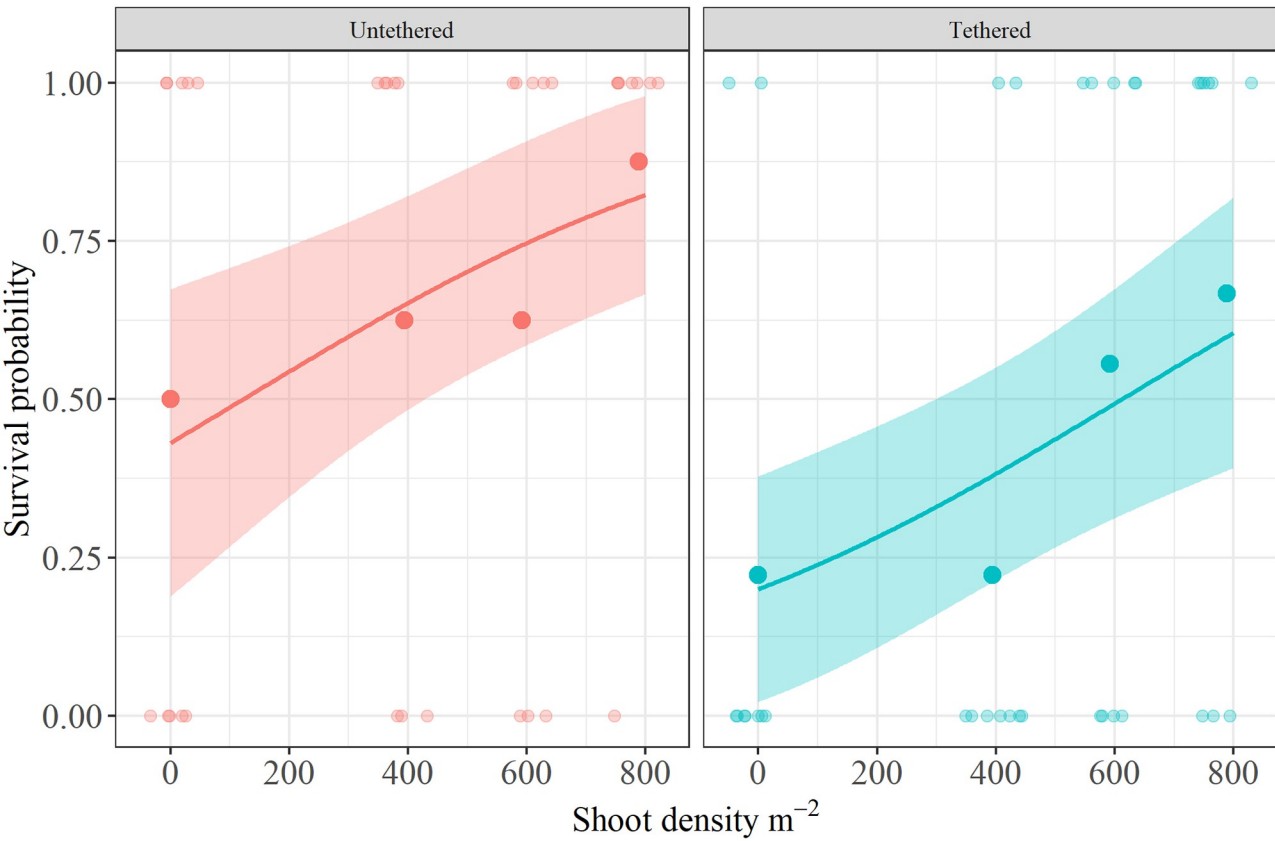

**Fig 3. Logistic regression mean conditional effects of shoot density and tethering status (tethered and free-swimming) on survival based on estimates from model $g_3$.** Black points in the center depict aggregated data: mean survival proportions across tethering treatments and shoot densities. Meanwhile colored points on the top and bottom of each plot depict observed data. Shaded regions denote 95% confidence bands.

support past research that did not test for the presence of treatment-specific bias in salt marshes [31, 32].

Survival was positively associated with shoot density, which is consistent with literature [25, 26, 49] and indicates that variation in structural complexity mediates survival of juvenile blue crabs as for other marine and estuarine juveniles [26, 50]. in concert with evidence of high abundance and growth of juvenile blue crabs in salt marshes [27, 30, 31, 51–54], our findings on survival support the conclusion that structurally complex *S. alterniflora* habitats enhance survival rates and promote high abundance of small juvenile blue crabs. Although flexible wooden dowels were a logical choice to simulate structurally complex *S. alterniflora* salt marsh habitat, we acknowledge that this simulated habitat may not be fully representative of an actual *S. alterniflora* salt marsh. Although we expect that the positive relationship between structural

**Table 4. Summary results of model $g_3$ coefficients: T denotes tethering and SD denotes shoot density.**

|  | Estimate | Std. Error | z value | P-value |
|---|---|---|---|---|
| intercept | -0.28 | 0.50 | -0.55 | 0.58 |
| SD | 2.22e-3 | 0.9e-3 | 2.48 | 0.01 |
| T | -1.11 | 0.53 | -2.09 | 0.04 |

complexity and survival observed both in this simulated salt marsh habitat as well as in other nursery habitats [25, 38] would extend to natural salt marsh habitat, we stress that inferences here would benefit from validation in a field setting.

The findings of this study represent important progress in understanding the function and value of alternative blue crab nursery habitats. Seagrass is considered the preferred nursery habitat for juvenile blue crabs [16]. However, seagrasses are threatened globally, and have declined markedly in recent decades [4, 55]. Moreover, the dominant seagrass species of the southern Chesapeake Bay, eelgrass *Z. marina*, is threatened due to increasing temperatures and poor water quality [56–58]. Although recent reductions in nutrient loads have led to recoveries in seagrass beds [59], future projections depict long term declines in *Z. marina* beds due to thermal stress, while projections of widgeon grass *Ruppia maritima* distributions remain uncertain and likely depend on further nutrient reduction [60]. Mounting evidence suggests salt marshes and certain unstructured sand habitats may be as valuable or more so at the population level due to their extensive areal cover [27, 30, 31]. Hence, it is critical that alternative nursery habitats, such as salt marshes, are protected for their contribution to blue crab populations.

## Future work

Although habitat-specific biases appear to be absent or negligible in salt marsh unstructured sand habitats, and seagrass habitats, there remains a need to validate tethering in other estuarine habitats. Juvenile blue crabs opportunistically utilize a broad suite of structurally complex and structurally simple habitats throughout their wide geographic range, such as coarse woody debris [50] and shallow detrital habitat [61, 62]. Comparisons of vital rates among all or most of the habitats that juveniles use is essential when assessing relative importance of any one particular habitat. Hence, future work examining treatment-specific biases among additional habitats utilized by juvenile blue crabs would be useful in determining when tethering is appropriate in comparing relative survival.

However, structural complexity is not the only characteristic of nursery habitats which moderates survival; environmental variables both among and within nurseries may influence juvenile survival within the seascape. Variation in environmental characteristics may promote or inhibit juvenile survival both within specific habitats (e.g. salt marsh; [63]) as well as among habitats (e.g. among salt marshes, seagrass beds, and oyster reefs; [64, 65]). For example, within- and among-habitat differences in water depth [66] and turbidity [67], can lead to vastly different survival rates for refuge-seeking juveniles. Moreover, environmental variables may synergistically interact with structurally complex habitat to provide superior refuge [31]. The absence of treatment-specific bias in tethering among salt marsh, seagrass beds, and unstructured sand habitats facilitates not only survival comparisons among the nominal habitats themselves, but also comparisons with respect to how these habitats and spatiotemporally covarying environmental variables influence juvenile blue crab survival within the seascape. This presents a promising avenue of future research both within Chesapeake Bay as well as throughout the wide geographical range of this species.

Finally, inferences in this study are limited by the use of a single predator species–adult blue crabs. While the ecology of juvenile blue crabs, and specifically the importance of larger conspecifics in influencing survival, made this choice a logical first step, several piscine predators– such as blue catfish *Ictalurus furcatus*, red drum *Sciaenops ocellatus*, and striped bass *Morone saxatilis*– also consume juvenile blue crabs at high rates [16, 46, 68]. Hence, future studies could expand inference via replicating these experiments with a piscine predator. in relation to inundation, additional experiments could simulate tidal dynamics rather than utilizing a static system.

## Supporting information

**S1 Data.**
(CSV)

## Acknowledgments

We thank Michael Seebo for mesocosm setup and advice on capturing, acclimating, and maintaining experimental animals; the Virginia Institute of Marine Science's NSF Research Experience for Undergraduates program, especially Prof. Rochelle Seitz and Dr. Grace Massey, for support and technical comments; and the University of Maryland Center for Environmental Science integration and Application Network media library for vector images used to create the conceptual diagram for Fig 1.

## Author Contributions

**Conceptualization:** Cole R. Miller, A. Challen Hyman, Romuald N. Lipcius.

**Data curation:** Cole R. Miller, A. Challen Hyman, Daniel H. Shi.

**Formal analysis:** Cole R. Miller, A. Challen Hyman.

**Funding acquisition:** Romuald N. Lipcius.

**Investigation:** Cole R. Miller, A. Challen Hyman, Daniel H. Shi.

**Methodology:** Cole R. Miller, A. Challen Hyman, Romuald N. Lipcius.

**Project administration:** Cole R. Miller, A. Challen Hyman, Daniel H. Shi.

**Resources:** Romuald N. Lipcius.

**Supervision:** A. Challen Hyman.

**Validation:** Cole R. Miller, A. Challen Hyman, Daniel H. Shi.

**Visualization:** Cole R. Miller, A. Challen Hyman, Daniel H. Shi.

**Writing – original draft:** Cole R. Miller, A. Challen Hyman, Daniel H. Shi.

**Writing – review & editing:** Cole R. Miller, A. Challen Hyman, Romuald N. Lipcius.

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
