## [Decision Letter · Decision Letter 0]

12 Mar 2023

PONE-D-23-02344Assessment of treatment-specific tethering survival bias for the juvenile blue crab Callinectes sapidus in a simulated salt marshPLOS ONE

Dear Dr. Miller,

Thank you for submitting your manuscript to PLOS ONE. After careful consideration, we feel that it has merit but does not fully meet PLOS ONE’s publication criteria as it currently stands. Therefore, we invite you to submit a revised version of the manuscript that addresses the points raised during the review process.

We look forward to receiving your revised manuscript.

Kind regards,

Goulven G Laruelle

Academic Editor

PLOS ONE

Journal Requirements:

   "The authors thank Michael Seebo for mesocosm setup and advice on capturing, acclimating, and maintaining experimental animals. Authors also acknowledge the Virginia Institute of Marine Science’s NSF Research Experience for Undergraduates program. Finally, authors acknowledge the University of Maryland Center for Environmental Science Integration and Application Network media library for vector images used to create the conceptual diagram for Figure 1. 

Preparation of this manuscript was funded by a Willard A. Van Engel Fellowship of the Virginia Institute of Marine Science, William & Mary, the NMFS-Sea Grant Joint Fellowship 2021 Program in Population and Ecosystem Dynamics, and the National Science Foundation (grant number NSF OCE 1659656 to R.D. Seitz)."

Additional Editor Comments:

Both reviewers find your study of potential interest for the community and are quite supportive in their overall evaluations but they also both point out the need for clarifications regarding the definition of the different hypothesis tested with your experiments and how they are connected to one another and to the experiments themselves. I believe that improving the description of the overall logical framework of this study will be beneficial to the manuscript and should be achieved by addressing the remarks of the reviewers.  

Also, I agree with reviewer #2 that the discussion currently remains a bit superficial and deserves to be expended a bit so I encourage the authors to follow some of the suggestions provided.

Reviewers' comments:

Reviewer's Responses to Questions

**Comments to the Author**

1. Is the manuscript technically sound, and do the data support the conclusions?

Reviewer #1: Partly

Reviewer #2: Yes

2. Has the statistical analysis been performed appropriately and rigorously? 

Reviewer #1: Yes

Reviewer #2: Yes

3. Have the authors made all data underlying the findings in their manuscript fully available?

Reviewer #1: Yes

Reviewer #2: Yes

4. Is the manuscript presented in an intelligible fashion and written in standard English?

Reviewer #1: Yes

Reviewer #2: Yes

5. Review Comments to the Author

Reviewer #1: This paper describes a set of experiments testing for interactions between tethering artifacts and treatments (artificial stem density) in mesocosm experiments. This is useful and important work, and it is refreshing to see papers properly grasping the implications of Peterson & Black 1994; such studies are often lacking, and such artifacts can invalidate the findings of tethering studies. The work seems sound, and the paper makes a useful contribution.

My main issues relate to:

1. the description of the experiments – more detail is needed and some work on the organization and flow is critically required to link the methods to the hypotheses and explain the experiments more clearly – see detailed comments below; and

2. some consideration of how well rigid wooden dowels at high density actually mimic Spartina stems in the field. My concern here is how reliable is the finding of lower mortality at higher stem density if the high density artificial dowels more effectively exclude large crabs than more flexible Spartina stems. I know they are pretty rigid vertically, but I strongly suspect a large crab could push through Spartina much more easily than through wooden dowels. This latter concern does not invalidate the substantive finding of no interaction between artifacts and treatment, but it does question the translation of findings about stem density into field scenarios.

3. L69, H5: the wording of this seems awkward or incorrect. If survival is inversely related to predator size, this means survival declines as predator size increases, yet the following explanation implies the opposite, i.e. survival increasing with larger predators because of lower efficiency in predation. Please clarify.

4. L72: should this be H5 not H15 for consistency?

5. L83: what is the relevance of citing a preprint for methods when it appears all the methods are described here? If they are not all described here, I would request that they should be, with no need to reference a preprint.

6. Fig 1: a photograph of a mesocosm would be nice. The figure seems oddly arranged. Why is the order of treatments different for the upper and lower panels?

7. The description of the experiments needs revising for clarity, and to relate them to the large number of hypotheses listed. For example, it is unclear which hypotheses are being tested by the experiments described at L114-124? What are these trials for, and how are they different to the trials described in the next paragraph?

8. L125: this explanation is also unclear. Why 4 juvenile crabs? I think I have figured it out (each trial involved 4 tethered and 4 untethered as per my comment below), but as currently written, the experiment descriptions are difficult to follow and to relate to the hypotheses posed.

9. Please provide more information about the tethering arrangement. Where was the tethered crab placed in the mesocosm and within the artificial stems? How much space was there around each pegboard in each mesocosm? Images of the actual tanks with the different densities would help. How far apart were the dowels in the high density treatment? Would this completely physically exclude the blue crab predators from entering the stems? It is also important to understand if both tethered and untethered crabs had access to both artificial stems and to open-water parts of each mesocosm.

10. L132: here too the description is confusing. How about saying something like “each trial involved a single replicate for each tethering (tethered or untethered) x shoot density (4 levels) combination, resulting in 8 tanks per trial.”

11. L140-141: these are unnecessary sentences. I suggest removing them. Maybe ok to mention using R for all analyses at the end of this section, but to say that all data were recoded digitally is redundant.

12. L151: Table 2 is cited before Table 1.

13. L166: please provide the specifics of how many replicates you ended up with for each treatment combination, e.g. in a table or just add the final sample size to Fig 1.

14. Fig 2 is redundant given the data in Table 1.

15. L200: do you mean interaction terms as well as predator size and prey size etc? This is a statement that just hangs and needs further explanation.

16. L206: “In addition,” is this statement not simply repeating what you already stated above as point 2 on L 203?

17. I understand the use of dowels as artificial Spartina stems, but I wonder how the rigidity of the dowels might impact your findings of the strong shoot-density effect? I imagine a large blue crab can move through real Spartina more easily than it can move through rigid dowls at high density. As per my earlier comment, more information is needed on the mesocosms, along with images of each treatment, and some discussion about the possibility that dowels at high densities might not be a realistic mimic of Spartina in the field, and therefore the conclusion that the experiments support the idea that higher density equals better refuge need to be tempered.

Reviewer #2: This study describes a set of experiments designed to determine if tethering is a valid method to understand the relative mortality (or survival) rates of juvenile blue crabs within salt marsh habitat. This is a useful question because the emergent vegetation within salt marshes are a somewhat under-recognized nursery habitat for blue crabs, particularly in regions without seagrass. The authors find that tethering did not induce a treatment-specific bias in survival of crabs across levels of marsh vegetation density, and is thus a useful methodology for future studies of crab survival in marshes.

The authors should clarify certain aspects of the Logical framework/hypotheses, but the larger workload will be to revise the discussion.

I hope that the authors find the following comments useful when revising the manuscript.

General comments

At some point prior to where it is currently written out, the authors should confirm that these are predation experiments. I recommend including this both in the Abstract and in the Logical framing section. This is important because survival could be measured over longer time scales and be, for example, part of density-dependence experiments (among others), so clarifying that these are short-term predation experiments is important.

Hypotheses: It’s unclear how H3 & H4 work together. In H3 you say that survival will be an additive function, while in H4 you say it may result in non-additive effects, indicative of treatment-specific bias. I don’t think you need each of these as separate hypotheses. I see the logical flow moving from H1 down to H6, but H3 and H4 need some clarification.

Relatedly, H5 seems to have an error, at least based on my understanding of “inversely”. That suggests a negative relationship, whereby prey survival would be lower at larger predator sizes. That is the opposite of what you then go on to describe in the justification for H5. Please revise and clarify.

Almost all of the first Discussion paragraph is redundant and could be removed, with the exception of the final sentence which should be revised and expanded (see comment below).

Much of the Discussion could be revised and expanded. It is short and mainly rehashes the Results section as opposed to critically discussing the mechanisms that lead to the results you found. One suggestion for an additional topic to include, considering this is a methods validation paper: what are the next 2-3 critical research questions that could be tackled using this technique, or what information could ultimately be provided to managers based on improved understanding of juvenile survival in marsh vegetation?

Specific comments

L3: “fisheries” don’t exhibit population dynamics. The stock/species that supports a fishery is what we refer to regarding population dynamics. The fishery is the social/commercial activity that utilizes the stock.

L39: alterniflora

L40: please define recruitment or revise this sentence to be more specific about the life stage/size you mean when you say “post-recruitment”

L44: does this specifically refer to tethering experiments? If not, please revise to clarify what is important about the interactive term. If yes, please clarify that you’re only talking about tethering.

L72: This says H1(sub)5, but I think you mean H5(sub)a and H5(sub)b, right?

L132: “trial”

L132: there is some redundancy in this methods section. Please revise to streamline and remove sentences with information that has already been provided.

Table order should be flipped, currently Table 2 is referenced prior to Table 1.

L180-181: replace dashes with () because currently it reads “g4 minus g3”

Fig 3: “points depict aggregated data;”- why not show each replicate and why use a semi colon in the caption? Were these logistic models fit to the aggregated data or the raw data? In my opinion it would be informative to see the raw data as opposed to (or at a minimum in addition to) these shoot density means values because it would be good to see how variable the effects of shoot density are on crab survival. Last, the legend is redundant because you already have titles on each panel.

L200: “all variables were informative” This sentence must need some clarification because all variables were not informative to explain variation in juvenile survivorship. You discarded temperature, salinity and DO at the outset, then your AICc-based model selection approach suggested that neither predator size nor prey size were useful. This sentence should be revised and equally important, expanded: tethering and shoot density were important, but why were the other variables not informative?

L219: there’s a ? in the list of references

L229-231 : seems like a bit of selective reference choice considering Lefcheck’s 2018 paper suggesting a huge recovery of seagrass in Chesapeake Bay. I suggest revising this sentence and toning down the dire nature of the predicted seagrass decline.

6. PLOS authors have the option to publish the peer review history of their article (what does this mean?). If published, this will include your full peer review and any attached files.

Reviewer #1: No

Reviewer #2: No

---

## [Author Response · Author response to Decision Letter 0]

2 Jun 2023

Dear Editor,

We thank both reviewers for the comments. Below is a description of how we addressed each recommendation and concern (bold font, with line numbers included). There was redundancy in reviewer concerns, and we referred to previous explanations in these instances. These revisions greatly improved the manuscript and figures.

Comments from Reviewer 1

1. The description of the experiments – more detail is needed and some work on the organization and flow is critically required to link the methods to the hypotheses and explain the experiments more clearly – see detailed comments below; and…

We acknowledge and agree with the reviewer that some additional detail was needed. We made adjustments using the points brought up throughout the revisions.

2. …some consideration of how well rigid wooden dowels at high density actually mimic Spartina stems in the field. My concern here is how reliable is the finding of lower mortality at higher stem density if the high density artificial dowels more effectively exclude large crabs than more flexible Spartina stems. I know they are pretty rigid vertically, but I strongly suspect a large crab could push through Spartina much more easily than through wooden dowels. This latter concern does not invalidate the substantive finding of no interaction between artifacts and treatment, but it does question the translation of findings about stem density into field scenarios.

We acknowledge the reviewer’s concerns; we addressed this point by explaining the following: First, the setup of the dowel within the board had the dowels loosely fit into their holes, giving them a small range of circular motion within their holes, allowing them to be displaced as if they were a plant. Personal observation from two of the authors saw the crabs successfully navigate the shoots. Moreover, the shoots were left within the tank for two weeks before the trials started, and thus the pegs were waterlogged and fouled, giving them additional flexibility along their entire length.

L104-109: “The wooden dowels employed were slightly smaller than the diameter of the holes within each pegboard, giving dowels a small range of motion within the holes and resembling flexibility observed in salt marsh vegetation. Adult crabs were also observed successfully navigating within the dowels. Additionally, dowels were left within the tank for two weeks before the experiment start, allowing them to be waterlogged and fouled, which increased additional flexibility.”

3. L69, H5: the wording of this seems awkward or incorrect. If survival is inversely related to predator size, this means survival declines as predator size increases, yet the following explanation implies the opposite, i.e. survival increasing with larger predators because of lower efficiency in predation. Please clarify.

We thank the reviewers for pointing out this issue and reworded the text for consistency by rephrasing the first point to match the second. 

L72-78: “H4: Survival is a function of tethering, shoot density, and predator size, such that survival is related to predator size because larger predators are restricted in movement, 

compared to smaller adults who can navigate through restricted spaces. As a result, large predators are less efficient than smaller predators when foraging for small juvenile crabs [32, 36– 38]. H4 considered both the effect of predator size within an additive model (H4a) as well as with the effect of predator size plus a tethering-shoot density interaction (i.e., treatment-specific bias; H4b)”

4. L72: should this be H5 not H15 for consistency?

We thank the reviewer for highlighting this typo and we have removed the erroneous l.

5. L83: what is the relevance of citing a preprint for methods when it appears all the methods are described here? If they are not all described here, I would request that they should be, with no need to reference a preprint.

We agree with the reviewer’s point and removed the citation as the citation is redundant and ensured that the methodology we used is fully outlined in the appropriate section.

6. Fig 1: a photograph of a mesocosm would be nice. The figure seems oddly arranged. Why is the order of treatments different for the upper and lower panels?

We acknowledge the reviewer’s comment and used an amended figure with a supporting photo of one of the tanks. See Figure 2.

7. The description of the experiments needs revising for clarity, and to relate them to the large number of hypotheses listed. For example, it is unclear which hypotheses are being tested by the experiments described at L114-124? What are these trials for, and how are they different to the trials described in the next paragraph?

We agree with the reviewer. We clarified that each trial is a replicate sample, and that analysis was done by testing the models on the trials. Each trial refers to a specific sample (or predator-prey event for a given tether-shoot density treatment). We combined both paragraphs to improve flow and clarity. However, we felt that the Data Analysis section clearly outlined the translation of the hypotheses into statistical models. 

8. L125: this explanation is also unclear. Why 4 juvenile crabs? I think I have figured it out (each trial involved 4 tethered and 4 untethered as per my comment below), but as currently written, the experiment descriptions are difficult to follow and to relate to the hypotheses posed.

We acknowledge the reviewer’s point and elaborated on the exact procedure with regards to the juveniles. We sampled 8 juveniles for each experiment and assigned one randomly to each tank, leading to four per treatment, but one per each tank. 

L147-151: “For each trial, one juvenile blue crab was randomly assigned to a single replicate in one of the two tethering treatments (tethered or untethered), with a given replicate consisting of one of four density treatments resulting in 8 tanks per trial (n = 2 tether treatments x 4 shoot density treatments = 8).”

9. Please provide more information about the tethering arrangement. Where was the tethered crab placed in the mesocosm and within the artificial stems? How much space was there around each pegboard in each mesocosm? Images of the actual tanks with the different densities would help. How far apart were the dowels in the high density treatment? Would this completely physically exclude the blue crab predators from entering the stems? It is also important to understand if both tethered and untethered crabs had access to both artificial stems and to open-water parts of each mesocosm.

We acknowledge the reviewer’s comments, and we added a paragraph to clarify the issues at the end of the design section. The paragraph outlines how the juveniles were added in the center of the tank near the standpipe (center). Shoot spacing was found with d ~ 1/(p^(1/dimensions)), giving measurements of 5.1, 4.14, and 3.6 cm for 388, 582, and 776 shoots per square meter density treatments, respectively. Both treatment prey items could reach the edge of the tank. 

L117-118: “Average spacing between the shoots was 5.1, 4.14, and 3.6 cm for the 388, 582, and 776 density treatments, respectively.”

L143-145: “A 25 cm thread of monofilament fishing line (4.5 kg test) was set into the glue [24, 26] with the length of the line allowing the juveniles to navigate the entirety of the tank.”

L156-158: “After recording its carapace width, a juvenile blue crab was placed in the center of each tank near the PVC standpipe regardless of treatment. The weight at the end of the tether was similarly placed at the center of the tank.”

10. L132: here too the description is confusing. How about saying something like “each trial involved a single replicate for each tethering (tethered or untethered) x shoot density (4 levels) combination, resulting in 8 tanks per trial.”

We agree with the reviewer’s comments and incorporated a similar sentence to reduce confusion. 

L147-151: “For each trial, one juvenile blue crab was randomly assigned to a single replicate in one of the two tethering treatments (tethered or untethered), with a given replicate consisting of one of four density treatments, resulting in 8 tanks per trial (n = 2 tether treatments x 4 shoot density treatments = 8).”

11. L140-141: these are unnecessary sentences. I suggest removing them. Maybe ok to mention using R for all analyses at the end of this section, but to say that all data were recoded digitally is redundant.

We agree with the reviewer’s point and removed the sentences.

12. L151: Table 2 is cited before Table 1.

We thank the reviewer for pointing out this typographical error and fixed the numbering. 

13. L166: please provide the specifics of how many replicates you ended up with for each treatment combination, e.g. in a table or just add the final sample size to Fig 1.

We acknowledge the reviewer’s point and amended the paper by adding a table listing the number of trails done in each replicate tank. Moreover, we caught an error regarding the number of replicates, which was 70 not 81. The manuscript and table now state the actual number. See Table 1.

14. Fig 2 is redundant given the data in Table 1.

We agree with the reviewer’s point and removed the old figure. 

15. L200: do you mean interaction terms as well as predator size and prey size etc? This is a statement that just hangs and needs further explanation.

We thank the reviewer for pointing out the incomplete statement. The remedy is outlined in reviewer 2’s comments, and we removed this paragraph for clarity. 

16. L206: “In addition,” is this statement not simply repeating what you already stated above as point 2 on L 203?

We agree with the reviewer’s comment and removed the redundant statement.

17. I understand the use of dowels as artificial Spartina stems, but I wonder how the rigidity of the dowels might impact your findings of the strong shoot-density effect? I imagine a large blue crab can move through real Spartina more easily than it can move through rigid dowls at high density. As per my earlier comment, more information is needed on the mesocosms, along with images of each treatment, and some discussion about the possibility that dowels at high densities might not be a realistic mimic of Spartina in the field, and therefore the conclusion that the experiments support the idea that higher density equals better refuge need to be tempered.

We agree and added sections regarding limitations of the experiment along with recommendations for future experiments with live shoots or more flexible, artificial shoots. We addressed this in the discussion section (see reviewer 2’s comments for more info).

L238-244: “Although flexible wooden dowels were a logical choice to simulate structurally complex S. alterniflora salt marsh habitat, we acknowledge that this simulated habitat may not be fully representative of an actual S. alterniflora salt marsh. Although we expect that the positive relationship between structural complexity and survival observed both in this simulated salt marsh habitat as well as in other nursery habitats [24, 40] would extend to natural salt marsh habitat, we stress that inferences here would benefit from validation in a field setting.”

 

Comments for Reviewer 2

General Comments

1. At some point prior to where it is currently written out, the authors should confirm that these are predation experiments. I recommend including this both in the Abstract and in the Logical framing section. This is important because survival could be measured over longer time scales and be, for example, part of density-dependence experiments (among others), so clarifying that these are short-term predation experiments is important.

We agree with the reviewer’s comment. We added a clarifying line in both sections about how this is a predation experiment. See Abstract. 

L58-60: “Under an Information Theoretic framework [33] we developed multiple alternative hypotheses (Hi) [34] for our short-term predation experiments. Herein we describe and justify the hypotheses and corresponding independent variables.”

2. Hypotheses: It’s unclear how H3 & H4 work together. In H3 you say that survival will be an additive function, while in H4 you say it may result in non-additive effects, indicative of treatment-specific bias. I don’t think you need each of these as separate hypotheses. I see the logical flow moving from H1 down to H6, but H3 and H4 need some clarification.

Relatedly, H5 seems to have an error, at least based on my understanding of “inversely”. That suggests a negative relationship, whereby prey survival would be lower at larger predator sizes. That is the opposite of what you then go on to describe in the justification for H5. Please revise and clarify.

We agree with the reviewer and altered the logical framework to be consistent with the other hypotheses, creating a theme where there is an additive hypothesis/model with an interactive sub-hypothesis/model while smoothing out inconsistencies with the numbering (including Table 2). 

L65-71: “H3: Survival is a function of tethering status and shoot density. The effects of shoot density and tethering status may affect survival additively (H3a). Alternatively, survival may be a function of tethering status, marsh shoot density, and their interaction ((H3b)). For example, juvenile blue crabs may have different escape strategies under differing habitat conditions, such as crypsis in structured habitat vs. escape in unstructured habitat [12, 35]. Tethering juveniles under different shoot densities may result in non-additive effects on survival (i.e., treatment-specific bias).”

3. Almost all of the first Discussion paragraph is redundant and could be removed, with the exception of the final sentence which should be revised and expanded (see comment below).

We agree with the reviewer and removed the paragraph and the final sentence.

4. Much of the Discussion could be revised and expanded. It is short and mainly rehashes the Results section as opposed to critically discussing the mechanisms that lead to the results you found. One suggestion for an additional topic to include, considering this is a methods validation paper: what are the next 2-3 critical research questions that could be tackled using this technique, or what information could ultimately be provided to managers based on improved understanding of juvenile survival in marsh vegetation?

We agree with the reviewer’s point. We added additional paragraphs to lengthen the discussion. We discussed future implications of the findings found here, specifically in reference to cross-habitat analyses with juvenile blue crabs, the effects of spatiotemporally varying environmental covariates, and change of predator. 

See Future Work Section (L259-292).

Specific Comments

5. L3: “fisheries” don’t exhibit population dynamics. The stock/species that supports a fishery is what we refer to regarding population dynamics. The fishery is the social/commercial activity that utilizes the stock.

We thank the reviewer for pointing out the factual error and fixed the term accordingly.

6. L39: alterniflora

We thank the reviewer for pointing out the grammatical error and we fixed the term accordingly. 

7. L40: please define recruitment or revise this sentence to be more specific about the life stage/size you mean when you say “post-recruitment”

We acknowledge the reviewer’s comment and changed the terminology to reflect the size class of juveniles used.

L41-43: “Previous studies estimating survival in small juveniles (i.e. 10-35 mm) have primarily focused on seagrass meadows, algae and unstructured sand habitats to infer nursery status [23–29].”

8. L44: does this specifically refer to tethering experiments? If not, please revise to clarify what is important about the interactive term. If yes, please clarify that you’re only talking about tethering.

We acknowledge the reviewer’s comments and addressed that this does in fact refer to tethering experiments only. 

9. L72: This says H1(sub)5, but I think you mean H5(sub)a and H5(sub)b, right?

We thank the reviewer for identifying the error and fixed it accordingly.

10. L132: “trial”

We thank the reviewer for pointing out the grammatical error and fixed it accordingly.

11. L132: there is some redundancy in this methods section. Please revise to streamline and remove sentences with information that has already been provided.

We acknowledge and agree with the reviewer’s point and we streamlined the section by removing redundant statements.

12. Table order should be flipped, currently Table 2 is referenced prior to Table 1.

We thank the reviewer for noticing this error and renumbered them properly.

13. L180-181: replace dashes with () because currently it reads “g4 minus g3”

We agree with the reviewer’s point and corrected the dashes to parentheses to smooth out the flow of the sentence.

14. Fig 3: “points depict aggregated data;”- why not show each replicate and why use a semi colon in the caption? Were these logistic models fit to the aggregated data or the raw data? In my opinion it would be informative to see the raw data as opposed to (or at a minimum in addition to) these shoot density means values because it would be good to see how variable the effects of shoot density are on crab survival. Last, the legend is redundant because you already have titles on each panel.

We agree with the reviewer’s comments, and we added the raw data to the graph. See Figure 3.

15. L200: “all variables were informative” This sentence must need some clarification because all variables were not informative to explain variation in juvenile survivorship. You discarded temperature, salinity and DO at the outset, then your AICc-based model selection approach suggested that neither predator size nor prey size were useful. This sentence should be revised and equally important, expanded: tethering and shoot density were important, but why were the other variables not informative?

We thank the reviewer for pointing out the error. We removed this sentence and elaborated on the specifics later in the discussion.

16. L219: there’s a ? in the list of references

We thank the reviewer for pointing out this citation error and we added the proper citation.

17. L229-231 : seems like a bit of selective reference choice considering Lefcheck’s 2018 paper suggesting a huge recovery of seagrass in Chesapeake Bay. I suggest revising this sentence and toning down the dire nature of the predicted seagrass decline.

We acknowledge the reviewer’s comment and clarified by explaining the details of seagrass we are referring to, specifically the decline in traditional Z. marina beds and the relative increase of widgeon grass Ruppia maritima.

L240-247: “However, seagrasses are threatened globally, and have declined markedly in recent decades [4, 57]. Moreover, the dominant seagrass species of the southern Chesapeake Bay, eelgrass Z. marina, is threatened due to increasing temperatures and poor water quality [58 – 60]. Although recent reductions in nutrient loads have led to recoveries in seagrass beds [61], future projections depict long term declines in Z. marina beds due to thermal stress, while projections of widgeon grass Ruppia maritima distributions remain uncertain and likely depend on further nutrient reduction [62].”

---

## [Decision Letter · Decision Letter 1]

19 Jul 2023

Assessment of treatment-specific tethering survival bias for the juvenile blue crab Callinectes sapidus in a simulated salt marsh

PONE-D-23-02344R1

Dear Dr. Miller,

We’re pleased to inform you that your manuscript has been judged scientifically suitable for publication and will be formally accepted for publication once it meets all outstanding technical requirements.

Kind regards,

Goulven G Laruelle

Academic Editor

PLOS ONE

Additional Editor Comments (optional):

Reviewers' comments:

Reviewer's Responses to Questions

**Comments to the Author**

1. If the authors have adequately addressed your comments raised in a previous round of review and you feel that this manuscript is now acceptable for publication, you may indicate that here to bypass the “Comments to the Author” section, enter your conflict of interest statement in the “Confidential to Editor” section, and submit your "Accept" recommendation.

Reviewer #1: All comments have been addressed

2. Is the manuscript technically sound, and do the data support the conclusions?

Reviewer #1: Yes

3. Has the statistical analysis been performed appropriately and rigorously? 

Reviewer #1: Yes

4. Have the authors made all data underlying the findings in their manuscript fully available?

Reviewer #1: Yes

5. Is the manuscript presented in an intelligible fashion and written in standard English?

Reviewer #1: Yes

6. Review Comments to the Author

Reviewer #1: The authors have done a good job responding to the reviewer comments, and I feel the paper is now much easier to follow. It is well laid out and makes some simple but important points, and makes an important contribution to the literature.

7. PLOS authors have the option to publish the peer review history of their article (what does this mean?). If published, this will include your full peer review and any attached files.

Reviewer #1: No

---

## [Editor Report · Acceptance letter]

24 Jul 2023

PONE-D-23-02344R1 

Assessment of treatment-specific tethering survival bias for the juvenile blue crab *Callinectes sapidus* in a simulated salt marsh 

Dear Dr. Miller:

I'm pleased to inform you that your manuscript has been deemed suitable for publication in PLOS ONE. Congratulations! Your manuscript is now with our production department. 

Kind regards, 

on behalf of

Dr. Goulven G Laruelle 

Academic Editor

PLOS ONE